# The Combined Effect of Acute Interval and Cognitive Training on Visual-Spatial Abilities in Women: Preliminary Insights for Health Promotion

**DOI:** 10.3390/ijerph22101524

**Published:** 2025-10-05

**Authors:** Christel Galvani, Sabrina Demarie, Ester Tommasini, Alessandro Antonietti, Thomas Zandonai, Paolo Bruseghini

**Affiliations:** 1Exercise & Sport Science Laboratory, Università Cattolica del Sacro Cuore, 20123 Milan, Italy; ester.tommasini@unicatt.it (E.T.); paolo.bruseghini@unicatt.it (P.B.); 2Department of Psychology, Università Cattolica del Sacro Cuore, 20123 Milan, Italy; alessandro.antonietti@unicatt.it; 3Department of Movement, Human and Health Sciences, University of Rome “Foro Italico”, 00135 Rome, Italy; sabrina.demarie@uniroma4.it; 4Department of Pharmacology, Paediatrics and Organic Chemistry, Miguel Hernández University of Elche, 03550 Sant Joan Alicante, Spain; tzandonai@umh.es; 5Addiction Science Laboratory, Department of Psychology and Cognitive Science, University of Trento, 38068 Rovereto, Italy

**Keywords:** high-intensity interval training, low-volume interval training, cognitive training, visual-spatial ability, health promotion

## Abstract

Different sports require elevated visual-spatial and related cognitive abilities, which are increasingly recognized as crucial not only for athletic performance but also for broader public health implications. Sex-related differences in these abilities have often been explained through both biological and sociocultural factors, with males traditionally described as having superior visual-spatial skills. However, fewer studies have investigated how targeted physical training can enhance these abilities in women. This study aimed to analyze the influence of two different cycling interval training exercises on visual-spatial ability in women. Seventy-two healthy, active, and young women engaged in (1) a High-Intensity Interval Training (HIIT) session followed by a cognitive training (CT); or (2) a Low-Volume Interval Training (LVIT) session followed by a CT; or (3) a cognitive (COG) session, consisting of listening to music followed by a CT; or (4) a control (CTRL) session, consisting of solely listening to music. Cognitive performance was assessed at baseline and after the training sessions using the Metzler and Shepard Test (MS), the Paper Folding and Cutting Test (PFC), and the Mental Rotation Test (MRT). No significant between-group differences were observed. However, in all groups the time to complete the PFC and MRT tests was significantly lower and the number of errors was significantly smaller for the PFC test in the post-test compared with the pre-test. These findings expand the current literature by demonstrating that interval training, whether high intensity or low volume, when combined with cognitive training, may improve certain aspects of visual-spatial cognitive performance in healthy, active, young women. These findings highlight the potential of combining structured exercise with cognitive challenges. Such interventions may promote cognitive health in women and contribute to long-term public health outcomes.

## 1. Introduction

In recent years, physical exercise has been widely associated with cognitive performance, with several studies showing that fitter individuals perform better in cognitive tasks [1,2]. Aerobic exercise appears to enhance various cognitive functions in humans, influencing brain plasticity, cognition, and overall well-being [3,4]. However, findings remain inconsistent, as methodologies differ depending on the type of cognitive task, the exercise performed, and the interval between exercise and cognitive assessment [5]. In fact, exercises of varying intensities exert different effects on cognitive function [6]. While moderate aerobic exercise improves cognitive abilities, evidence suggests that anaerobic activity performed close to or at 100% of maximal oxygen consumption (V′O_2_max) may reduce these abilities [7]. A substantial body of literature has focused on the effects of a single session of exercise on cognitive performance. Reviews indicate that exercise enhances performance on cognitive tasks conducted during or after the session, with higher intensities producing longer-lasting effects [8]. Although it has been shown that combining mental and physical training is particularly effective compared to performing either activity alone in longitudinal studies [9], no studies have investigated the impact of an acute bout of combined physical training and cognitive training (CT) on cognition.

Recently, interval training (IT) has gained popularity as a moderate-to-high intensity exercise strategy among both the general population and athletes. IT is characterized by repeated bouts of cyclic exercise alternated with periods of rest, which allow the activity to be sustained for a longer overall duration [10], compared to continuous exercise performed at the same intensity. Adjustments in IT intensity and duration have led to different training approaches, among which High-Intensity Interval Training (HIIT) and Low-Volume Interval Training (LVIT) are the most common. Both HIIT and LVIT involve relatively short, repetitive bouts of training performed in an “all-out” manner at an intensity capable of eliciting peak oxygen consumption (V′O_2_peak), interspersed with recovery periods of rest or low-intensity exercise [11,12]. These training modalities provide numerous health benefits, particularly for cardiometabolic parameters and physical fitness [13,14]. In addition, they can achieve substantial exercise effects in a shorter time compared to traditional aerobic training. However, the benefits of IT on cognitive function, especially on complex skills such as visual-spatial ability, remain unclear. Moreover, most studies in this field have been conducted on men, with relatively few focusing on female cohorts [10].

Spatial cognition enables environmental navigation and orientation, allowing humans to perceive spatial relationships, estimate distances, and mentally manipulate objects, skills essential for daily activities and complex tasks. Visual-spatial abilities—including spatial perception, visualization, and mental rotation [15]—depend on the brain’s capacity to integrate sensory information and create internal spatial maps, facilitating precise movement planning and execution. Athletes face intensive demands on visual-spatial processing across sport disciplines. Regular physical activity refines motor skills while strengthening cognitive functions, including spatial awareness, attention, and reaction time [16,17]. Research in badminton [18] and table tennis [19] shows that athletes consistently outperform non-athletes in tasks requiring rapid spatial judgment and decision making. Athletes also exhibit superior mental rotation abilities, suggesting that intensive training in dynamic environments enhances spatial information processing [20]. Athletic expertise manifests through sport-specific adaptations. Soccer players with high expertise demonstrate accelerated recognition of soccer-specific poses in left–right decisions, reflecting enhanced egocentric mental rotation. Conversely, gymnasts excel at recognizing poses rotated around multiple axes and show superior performance with novel cube figures and hand drawings, while letter rotation remains comparable. These findings indicate that sport-specific practice selectively enhances visual-spatial skills: soccer favoring egocentric transformations or gymnastics promoting object-based transformations [21]. Spatial cognition relies on a distributed neural network: the parietal cortex integrates sensory inputs for spatial representations, the hippocampus supports spatial memory, and the prefrontal cortex orchestrates planning and decision-making. In athletes, these regions demonstrate enhanced activation and connectivity, reflecting neural adaptations from intensive training and cognitive demands, underscoring the connection between motor skills and cognition [22].

Tests of mental rotation have been previously shown to be one of the most prominent sex differences in cognitive psychology, marked by a large male performance advantage [23]. Literature has surfaced suggesting that purported sex differences can be attributed to genetic, social, cognitive, and performance factors [24]. It is well known that females are relatively less physically active than males [25]. Young females are at a particularly high risk of being sedentary and physically inactive, and this inactivity behavior could diminish cognitive function, increase stress, and anxious emotions [26]. Moreover, a significant under-representation of women included in sport and exercise medicine research studies still remains [27]. To the best of our knowledge, no studies have investigated visual-spatial abilities with IT in females and no studies have investigated the combination of an acute IT and CT bout on visual-spatial abilities.

The purpose of this study was to investigate the influence of one IT and CT session on visual-spatial performance in healthy young women. We thus hypothesized that the combination of HIIT or LVIT and CT would be associated with enhanced short-term effects on visual-spatial tests. This effect may positively impact many sports by compensating for the decline in cognitive performance during high-intensity exercise.

## 2. Materials and Methods

### 2.1. Subjects

Seventy-two healthy active young women volunteers were recruited by local advertisements at the Exercise and Sport Science Courses of the Università Cattolica del Sacro Cuore, Milan. All participants were informed about the procedures and risks of the study and informed consent was obtained from all subjects involved in the study. Each participant was given an information sheet indicating the behavior to adopt during the experimental protocol. All participants met the criteria for inclusion: no previous history of orthopedic or musculoskeletal injuries in the past 2 years, an age range between 20 and 30 years, and a BMI lower than 25 kg∙m^−2^. After a preliminary medical examination, to evaluate possible pathological responses to exercise, volunteers were randomized into 4 groups: HIIT, LVIT, Cognitive Training group (COG), or Control group (CTRL). Exclusion criteria were the presence (self-reported) of no medically stable acute or chronic disease, neurological disorders, other motor or cognitive restrictions which could influence the regular outcome of the study, or current pharmacotherapy which could alter the subjects’ metabolic and exertional responses. Participants’ characteristics are described in Table 1.

In order to control the impact of the menstrual cycle phase on perceived physical and cognitive performance outcomes, the specific moments of the menstrual cycle—i.e., follicular phase (FP), luteal phase (LP), and menstrual phase (MP)—and oral contraceptive use (OC) were reported by each subject. Furthermore, participants self-reported experiencing stressful events (e.g., being stuck in traffic when already late or arguing with a family member or friend) just prior to each training session. The study was conducted in accordance with the current Declaration of Helsinki guidelines and was approved by the Ethical Commission (approval date 20 November 2013).

### 2.2. Experimental Design

The study design consisted of a randomized, controlled, between-subject design. Participants visited the lab on 2 different occasions, separated by at least 48 h. Environmental conditions in the lab were controlled and maintained across all visits (temperature of 18–20 °C, and humidity of 50%). Participants were asked to eat breakfast or lunch no more than 2–3 h before each session, to maintain a stable caloric intake, and not to participate in any physical exercise in the weeks in which they were occupied in the experimental protocol. Specifically, participants were advised to avoid moderate or vigorous physical activity during the 48 h prior to each assessment session. Furthermore, participants were instructed to refrain from consuming coffee, alcohol, drugs, and tobacco in the 24 h prior to the workout, and to have dinner and go to bed at the same time each day, ensuring at least 8 h of sleep [28]. During the first day, anthropometric measurements, physical activity level, and cognitive performance were assessed. On the second day, the experimental sessions were carried out as follows:(1)HIIT session followed by cognitive training;(2)LVIT session followed by cognitive training;(3)COG session, consisting of the same time period as the HIIT session (1), with listening to music followed by cognitive training;(4)CTRL session, consisting of the same time period as the LVIT session (2), with listening to music.

Each group was assigned to a single training protocol.

Cognitive performance was evaluated at baseline and post each experimental session.

### 2.3. Procedures

#### 2.3.1. Anthropometry and Physical Activity

Body mass and stature were taken at the nearest 0.1 kg and 0.01 m, respectively, with a scale and a stadiometer (Vandoni, Salus, Milan, Italy). Body mass index (BMI, kg·m^−2^) was calculated as body mass divided by the square of body height.

Physical activity rating (PAR, score 0–7) was investigated through the Jackson questionnaire [29]: a lower PAR score, such as 0 or 1, indicates a sedentary lifestyle, while a higher score (e.g., 6 or 7) suggests a high level of physical activity. This tool takes into account the physical activity performed regularly on average in a week, assessed by intensity and volume. In particular, the activities considered are (i) at vigorous intensity (e.g., 24 km run per week); (ii) at moderate intensity (e.g., up to an hour or more by practicing horse riding or rhythmic gymnastics); (iii) at low intensity, performed occasionally (e.g., walking or climbing stairs); (iv) no activity (includes the use of the car and all the habits of a sedentary lifestyle).

#### 2.3.2. Cognitive Performance

##### Metzler and Shepard Test (MS)

The MS test is a mental rotation task consisting of 15 items [30]. The test was administered to the subjects in digital format without time limits. Two three-dimensional figures placed side by side were presented on the screen; the subject compared them and indicated if they were the same object presented in two different angles or two different objects. The level of difficulty of the items is given by the presence of mirror objects, the structure of the object, and the relative degree of orientation. The latter factor significantly influences the response times when the subject makes a mental rotation of the object both along the vertical-horizontal and in the depth axes.

##### Paper Folding and Cutting Test (PFC)

The PFC test was a folding and overturning task. It was a sub-test of the Stanford–Binet intelligence scale for the evaluation of visual-spatial ability [31]. This computerized test was administered without a time limit. It consists of 15 items, each item entailing 5 stimuli. The stimulus (target) to the left of the screen represents a sheet of paper that is folded and cut in different ways. To the right of the target stimulus, four alternatives are shown, represented by four “open” sheets with different folds and cuts. The subject must choose the correct alternative. The execution of the task requires active manipulation of the reference figure, specifically of the overturning operations. The difficulty level of the items grows linearly as the number of folds and cuts increases.

##### Mental Rotation Test (MRT)

The MRT has been used to measure visual-spatial abilities, specifically the ability to rotate mental images [32]. This test consists of 24 items, each of which represents a three-dimensional object in two dimensions that must be compared with the others. Each item consists of five figures: the reference stimulus placed on the left and four figures on the right, two of which are distractors (i.e., figures that do not correspond to the correctly rotated target stimulus). For each of the 24 items, the subject has to identify the two figures among the four proposals that correspond to the reference stimulus correctly rotated. To avoid subjects being given the same item on two consecutive days, it was decided to divide the test into two parts: the first part (from item 1 to item 10) was administered during the first day and the second (from item 11 to item 20) was administered during the post-test phase. The time limit available to the subjects to carry out the task has been re-proportioned from 8 to 4 min per tranche [33,34].

#### 2.3.3. Experimental Treatment

##### High-Intensity Interval Training (HIIT)

The HIIT protocol consisted of four 30 s all-out Wingate repetitions (W1: first repetition, W2: second, W3: third, W4: fourth) performed on a cycle ergometer (Excalibur Sport, Lode, Germany). Each Wingate sprint was interspersed with a 4-min recovery interval at a low workload (approximately 50–30 W) [35]. The training session was preceded by 10 min of an active warm-up and by 5 min of a specific warm-up on a cycle ergometer at 50 W. The participant was encouraged to pedal as fast as possible against a fixed resistance of 0.75 Nm/kg body mass. The entire supervised training session lasted about 25 min, including the post training cool-down phase. During HIIT, heart rate (HR) (Polar Electro Oy, Kempele, Finland) and workload (W) were monitored. At the end of exercise, the subjects were asked to rate fatigue, according to the Borg CR-10 scale [36].

##### Low Volume Interval Training (LVIT)

LVIT consisted of three 20 s all-out Wingate (W1, W2, W3) cycling (Excalibur Sport, Lode, Germania) interspersed by 2 min recovery intervals at about 50 W [37]. Each series was preceded by 10 min of an active warm-up and by 2 min of a specific warm-up on a cycle ergometer at 50 W. The participant was encouraged to pedal as fast as possible against a fixed resistance of 0.50 Nm/kg body mass. The entire supervised training session ended in about 20 min, including the post training cool-down phase (3 min at 50 W). During LVIT, HR and power were monitored. At the end of exercise, ratings of perceived exertion (RPE) were recorded [36].

##### Cognitive Training

Cognitive training focused on orientation and spatial visualization. Accordingly, mental image manipulation exercises were proposed. These exercises were similar to the twisting and bending exercises used to assess subjects on the pre- and post-test [38]. Cognitive training was composed of three types of exercises: reversal of a chessboard, reassembling a solid figure, and visualization and spatial orientation. The whole training was completed in about 20 min. The structure of the training is described in Table 2. In brief, the cognitive training was administered in Italian using paper-based materials. Participants were instructed to indicate the correct response for each item with a pen (see also Appendix A). At the beginning of the training no strategies were suggested, either verbally or visually, and the experimenter provided no performance-related feedback during the CT sessions, thereby minimizing potential instructional confounds.

### 2.4. Statistics

The results were expressed as means and standard deviation (SD). The Kolmogorov–Smirnov test was applied to test the normality of the data. In order to analyze the influence of an acute bout of HIIT or of LVIT on visual-spatial abilities, a multivariate analysis of variance (MANOVA) was used. Groups (HIIT, LVIT, COG, CTRL) and time (pre, post) were considered as independent variables, whereas the collected measures, such as response number and response time, were treated as dependent variables. Subsequently, a two-way analysis of variance (ANOVA) was used, followed by Fisher’s LSD post hoc test. Statistical Power estimates were calculated for both significant and non-significant outcomes. The models were adjusted for the phase of the menstrual cycle, oral contraceptive use, and exposure to a stressful event. Furthermore, in order to control the physiological differences elicited by an acute bout of HIIT compared to an acute bout of LVIT, a one-way analysis of variance (ANOVA) was used. Differences were accepted as significant if *p* < 0.05. Statistical analysis was performed using StatView, version 5.0.1.

## 3. Results

### 3.1. Main Characteristics of the Subjects

Participants included healthy women who were (a) 12% in FP, 67% in LP, and 21% in MP and (b) 47% not taking any hormonal contraceptives. Moreover, 66% of the subjects reported no stressful events. No significant differences were detected between groups for anthropometric parameters (Table 1).

### 3.2. Physiological, Mechanical, and Fatigue Data During Training Sessions

During HIIT and LVIT the mechanical power decreased and RPE increased throughout the exercise, indicating an increased fatigue state in both groups (Table 3). Consistently, a significant (*p* < 0.001) decrease in mean power output was found in the HIIT and VLIT groups, with a significantly greater extent in HIIT (*p* < 0.05). Similarly, a significant (*p* < 0.0001) rise in RPE was found in both groups.

The mean HR values recorded during HIIT and LVIT, as well as during cognitive training, are reported in the Appendix A.

### 3.3. Cognitive Performance Data

No significant between-group differences were observed either in the pre- or in post-tests. No statistically significant differences were found between pre- and post-test mean scores in the MS test, despite minor variations (Figure 1).

The time to complete the PFC and MRT tests was significantly (*p* < 0.0001, Power = 1.0; *p* < 0.05, Power = 0.661, respectively) lower and the number of errors significantly (*p* < 0.0001, Power = 1.0) decreased for the PFC test in the post-test compared with the pre-test (Figure 2 and Figure 3). In contrast, significant differences were observed neither in the MS test for response number (ns, Power = 0.080) or response time (ns, Power = 0.438), nor in the MRT for the number of correct responses (ns, Power = 0.050).

## 4. Discussion

The purpose of this study was to examine the effect of a single session of High-Intensity Interval Training or Low-Volume Interval Training combined with cognitive training on visual-spatial performance in young adult women. Improvements in certain visual-spatial abilities were observed across all groups, indicating that cognitive training alone can enhance performance and that incorporating interval training—regardless of intensity—does not detrimentally impact cognitive outcomes. This demonstrates that physical exercise can be combined with cognitive interventions without negative effects.

The study involved a sample of young, healthy, and physically active women. This choice is particularly significant, as it distinguishes the present research from much of the existing literature, which has focused primarily on young adult male soccer players, middle-aged individuals, or older adults [39,40,41]. By directing attention to women, who are often underrepresented in studies on visual-spatial abilities, this work contributes to addressing an important gap in the scientific field. Considering that women generally show lower performance in mental rotation tasks, it becomes especially relevant to investigate how their visual-spatial skills respond to cognitive training combined with acute exercise. From a public health perspective, these insights are crucial, since enhancing cognitive and perceptual abilities through accessible training strategies can support not only athletic development but also lifelong cognitive well-being and resilience in women [42].

Our study evaluated visual-spatial abilities after IT protocols followed by cognitive training. Many studies have adopted tasks to assess reaction time and visual recognition [43]; others have examined the effects of acute exercise on executive function, selective attention, and short-term memory [5,44,45,46,47]. Spatial visualization abilities still seem underexplored in this field of research. In fact, few studies have evaluated these functions after acute exercise [33,39]. Furthermore, studies using an acute exercise training protocol are very heterogeneous. Most of them have adopted aerobic exercises of various durations but lasting more than 20 min [43,48,49,50,51,52]. To date, only one author has used HIIT to evaluate its effects on cognitive performance in healthy middle-aged adults, demonstrating that the exercise protocol performed seemed to be able to improve the performance on a selective attention task [39]. The results were apparently in contrast with the U-shaped hypothesis; an inverted U-relationship was in fact observed between the cognitive performance and the exercise workload, suggesting that the intensity of the exercise is a key factor in the regulation of exercise-induced cognitive responses. According to this theory, acute, moderate intensity, and continuous aerobic exercise can promote an increase in cognitive performance. Conversely, high intensity exercise may decrease cognitive performance [51]. In our study, IT resulted in a significant improvement in the PFC and MRT output data, confirming the data published by Alves et al. [39], who observed an improvement in selective attention immediately after a HIIT session, and contrasting with the inverted-U theory, which predicts a worsening of cognitive performance [51]. Furthermore, the improvement in cognitive performance associated with the onset of fatigue, assessed through RPE and power output values, does not support the hypothesis of Alves et al. [39], who argued that exercise-induced fatigue may play an important role in the cognitive responses to an exercise bout. On the contrary, our results clearly indicate that the onset of acute fatigue—marked by both a reduction in power output and an increase in RPE during exercise—does not appear to negatively affect cognitive performance, suggesting that cognitive abilities can remain stable even under substantial physiological stress. The observed improvements in cognitive performance across both groups may reflect the development of novel problem-solving strategies induced by cognitive training, suggesting not only task-specific gains but also potential transfer effects to broader cognitive processes. The complementary and modulating effects of CT on HIIT and LVIT significantly reduced both the time required to complete the PFC and MRT tests and the error count on the PFC test.

Based on the literature review, the present study seems to be the first to investigate the effect of the combination of an exhaustive work protocol followed by cognitive training on visual-spatial abilities. Furthermore, compared with Alves et al. and Lemmink & Visscher [39,53], the present study considered, as covariates, data relating to the menstrual cycle, oral contraceptives, and any stressful events that may have occurred, as they can have an influence on physical and cognitive performance [54,55,56].

Recent studies have demonstrated the usefulness of physical activity in improving cognitive processes. Cognitive skills can be improved with targeted, individualized workouts that induce better neuronal efficiency by combining physical and cognitive training [57]. In particular, exercises that simultaneously engage motor and cognitive demands appear to enhance synaptic plasticity, attentional control, and executive functioning. The study of the influence of acute physical exercise on cognitive processes could be an interesting approach for the development of new methods for performance optimization. The importance of the link between sports performance and cognitive functioning has already been highlighted by both coaches and athletes, especially in sports where a large amount of information must be processed in a very short time [58]. Understanding the variables that influence the interaction between cognitive processes and acute physical exercise—such as exercise intensity, duration, and type—could allow the design of more effective training protocols. By tailoring workouts to the specific cognitive and physical demands of a sport, it may be possible to improve reaction times, decision-making speed, and situational awareness, ultimately optimizing overall sports performance and reducing errors during high-pressure scenarios. In this regard, this apparent dissociation observed in the MRT—faster responses without a significant increase in accuracy—may reflect a speed–accuracy trade-off (SAT), a well-documented phenomenon in cognitive psychology in which response speed is prioritized at the expense of accuracy. As highlighted by Guo et al. [59], repeated task exposure and increasing familiarity with stimuli can reduce cognitive load, thereby facilitating quicker responses without necessarily improving correctness, a pattern that may have important implications for sports activities requiring rapid decision-making under time pressure.

According to the literature, it seems that women may respond “less” to interval training [10,60]. Therefore, it is important for future research to assess whether modifications to the interval training stimulus can enhance responses in women. Moreover, it would be valuable to examine potential differences between women engaged in different sports while also considering body composition and the possible integration of a nutritional intervention. To better isolate the effects of the interventions and account for variability, future studies should include a true passive control group and, additionally, consider using larger sample sizes. A larger sample size may be necessary to reliably detect modest enhancements in cognitive performance, particularly regarding the MS test, where minor improvements were observed in our study but failed to achieve statistical significance. Manipulation of interval exercise prescription variables, such as exercise intensity, duration, frequency, and work-to-recovery ratio, represents possible strategies to help women respond more effectively to interval training. Finally, individual physiological differences, hormonal fluctuations, and training history should be considered when designing interval programs, as these factors may influence responsiveness and optimize the benefits of training interventions for female participants.

Despite the results observed, the interpretations are subject to several limitations that warrant caution. First, the lack of diverse and more valid physiological parameters may have constrained the evaluation of fatigue. Second, the use of a cycle ergometer to perform HIIT and LVIT, while ensuring safety given the variety of sports practiced by our participants (e.g., volleyball, soccer, swimming, gymnastics, track and field, dance), likely did not allow for a fully personalized assessment of performance across all individuals. Furthermore, the study may have been influenced by significant practice effects due to the relatively short interval between pre- and post-testing and the considerable similarity between the CT tasks and the assessment measures. The absence of a true control group, which did not engage in any form of training, limits the ability to establish an accurate baseline for these practice effects.

From a practical standpoint, these findings suggest that cognitive training, when combined with either HIIT or LVIT, can effectively enhance visual-spatial performance in active young women, highlighting the central role of cognitive interventions and supporting the integration of time-efficient, combined physical and cognitive approaches in public health strategies aimed at improving cognitive function in women. Specifically, such protocols can be implemented in cognitive health promotion, workplace wellness programs, and sports training. In public and occupational settings, brief interval training combined with cognitive tasks may help improve mental focus, spatial awareness, and overall cognitive resilience. In sports contexts, this strategy may contribute to better performance, reflecting the multifaceted nature of athletic demands.

## 5. Conclusions

High cognitive function is increasingly recognized as a key determinant of public health, as it enables individuals to preserve mental and physical well-being, make informed decisions, and interact effectively with their environment, thereby reducing the risk of chronic illnesses and age-related cognitive decline [42,61]. Previous research has shown that cognitive performance is associated with both mental and physical health, and its predictive value may even strengthen from midlife to early old age, possibly influenced by socioeconomic status and life transitions such as retirement. It is now widely recognized that acute aerobic exercise can improve specific cognitive functions, such as short-term memory, selective attention, processing speed, and aspects of inhibitory control. These beneficial effects are often attributed to enhanced cerebral blood flow, increased arousal, and the release of neurotrophic factors, such as BDNF, which support synaptic plasticity and overall brain function [5,7,45,47,51,62]. However, an inverted U-shaped relationship has been observed between cognitive activation and exercise workload, suggesting a possible intensity threshold beyond which cognitive performance may be impaired and not further enhanced by training [62,63]. The negative effects of training at intensities above the anaerobic threshold on cognitive performance have been reported in some, but not all, assessment tasks. Apparently, in contrast to this theory, in the present study no detrimental effects on cognitive performance were observed following an acute bout of interval training, which, when integrated with cognitive training, appears not to interfere with cognitive functioning and may therefore represent a compatible strategy within combined intervention approaches. Improvements in administered tasks were noted, corroborating previous data where above-threshold training did not create any interference with cognitive performance [39,53,64]. These findings suggest that the relationship between exercise intensity and cognitive enhancement may be more complex than previously thought, potentially influenced by factors such as task specificity, participant fitness level, and exercise duration.

As highlighted by Fransen (2024), the relationship between cognitive training and physical training is not yet fully understood [65]. While some evidence suggests potential benefits for sport-related outcomes, the findings remain inconsistent, and it is unclear how effectively this training translates into meaningful improvements in real-world performance. Outcomes may be influenced by factors such as individual differences, training parameters, and the specific demands of each sport. Consequently, the precise role of cognitive training in supporting physical activity remains to be determined, highlighting the need for further research in applied settings.

## Figures and Tables

**Figure 1 ijerph-22-01524-f001:**
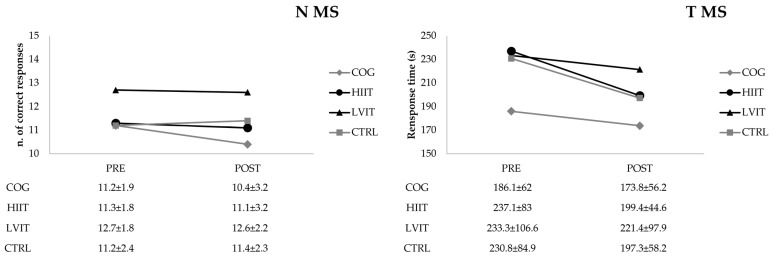
Number of responses and response time of Metzler and Shepard Test (MS); N: number; T: time; COG: Cognitive Training group; HIIT: High-Intensity Interval Training group; LVIT: Low-Volume Interval Training group; CTRL: Control group.

**Figure 2 ijerph-22-01524-f002:**
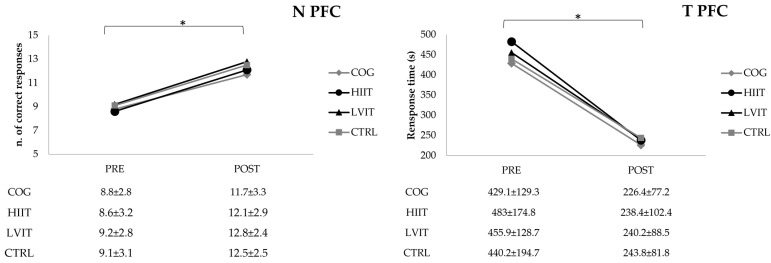
Number of responses and response time of Paper Folding and Cutting Test (PFC). N: number; T: time; COG: Cognitive Training group; HIIT: High-Intensity Interval Training group; LVIT: Low-Volume Interval Training group; CTRL: Control group. * significant difference pre-test vs. post-test (*p* < 0.0001).

**Figure 3 ijerph-22-01524-f003:**
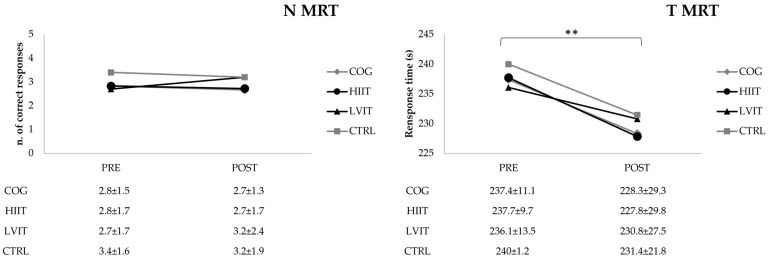
Number of responses and response time of Mental Rotation Test (MRT). N: number; T: time; COG: Cognitive Training group; HIIT: High-Intensity Interval Training group; LVIT: Low-Volume Interval Training group; CTRL: Control group. ** significant difference pre-test vs. post-test (*p* < 0.05).

**Table 1 ijerph-22-01524-t001:** Anthropometric and physiological parameters.

	TOT (N72)	HIIT (N18)	LVIT (N18)	COG (N18)	CTRL (N18)
Age (yrs)	21.8 ± 1.2	21.3 ± 0.7	21.6 ± 0.9	22.5 ± 1.7	21.7 ± 1.1
Height (m)	1.65 ± 0.05	1.66 ± 0.06	1.66 ± 0.06	1.64 ± 0.05	1.65 ± 0.04
Weight (kg)	58.5 ± 6.4	58.7 ± 5.6	58.5 ± 6.2	58.1 ± 6.8	58.6 ± 7.3
BMI (kg·m^−2^)	21.4 ± 2.0	21.4 ± 1.6	21.3 ± 2.3	21.4 ± 1.8	21.4 ± 2.3
Score PAR	4.6 ± 1.6	5.0 ± 1.0	4.3 ± 1.7	4.6 ± 2.1	4.5 ± 1.2

HIIT: High-Intensity Interval Training group; LVIT: Low-Volume Interval Training group; COG: Cognitive Training group; CTRL: Control group; PAR: Physical activity rating.

**Table 2 ijerph-22-01524-t002:** Training structure: types of spatial visualization exercises.

Reversal	N. Item	Rotation	N. Item	Visualization and Spatial Orientation	N. Item
Chessboards	3	Reconstruction of cubes A	2	Change of perspective A	3
		Reconstruction of cubes B	3	Change of perspective B	2
		Comparison of cubes	3		

**Table 3 ijerph-22-01524-t003:** Mechanical and RPE data during physical training.

	Max Power (W)		Mean Power (W)		RPE	
	HIIT	LVIT	HIIT	LVIT	HIIT	LVIT
W1	539.4 ± 111.5	514.5 ± 73.2	396.6 ± 54.5	388.9 ± 44.8	4.1 ± 1.6	3.1 ± 1.1
W2	555.8 ± 93.1	499.6 ± 71.6	379.3 ± 49.7	372.8 ± 51.6	5.1 ± 1.6	4.2 ± 1.6
W3	513.5 ± 115.8	499.6 ± 84.5	331.0 ± 58.6	356.7 ± 60.7	6.4 ± 1.7	5.6 ± 2.1
W4	525.5 ± 111.5		324.4 ± 59.5		7.7 ± 1.6	

HIIT: High-Intensity Interval Training group; LVIT: Low-Volume Interval Training group; W1: first all-out Wingate repetition; W2: second all-out Wingate repetition; W3: third all-out Wingate repetition W4: fourth all-out Wingate repetition.

## Data Availability

The raw data supporting the conclusions of this article will be made available by the corresponding author on request.

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
