# Peer review of "The Combined Effect of Acute Interval and Cognitive Training on Visual-Spatial Abilities in Women: Preliminary Insights for Health Promotion"

_ijerph, 2025, doi:10.3390/ijerph22101524_

Round 1
Reviewer 1 Report
Comments and Suggestions for Authors
The manuscript presents a novel and timely contribution to the intersection of physical training, cognitive enhancement, and gender-specific health promotion. The study's strengths, including its experimental design, use of multiple validated cognitive tests, and focus on a female cohort, are commendable. However, there are several areas where clarity, depth, and linguistic precision can be improved to enhance the overall impact of the manuscript.
Strengths
-
Relevance and Novelty: This topic is highly relevant, given the underrepresentation of women in exercise cognition research. Your focus on acute interventions and their effects on visual-spatial cognition in women addresses a clear gap in the literature. As you note, ‘no studies have investigated visual-spatial abilities with IT in females’ (p. 3), which underscores the originality of your approach.
-
Experimental Design: The randomized controlled structure with four distinct groups (HIIT, LVIT, COG, and CTRL) allows for comparative analysis and control of confounding factors. Additionally, adjusting for menstrual phase, contraceptive use, and stress exposure demonstrates attention to biological and environmental influences on cognition (p. 5, lines 139–145).
-
Methodological Rigor: The cognitive tests used—Metzler and Shepard (MS), Paper Folding and Cutting (PFC), and Mental Rotation Test (MRT)—are well-established tools in cognitive psychology. The combination of these tests offers a multifaceted evaluation of visuospatial skills.
Areas for Improvement
-
Clarity and Language Precision
-
Throughout this manuscript, there are numerous instances of awkward or incorrect phrasing that may hinder reader comprehension. For instance, in the abstract (p. 1), the phrase “can effectively enhance visual-spatial cognitive performance in healthy, active, young women” could be more precise. Consider revising to “may improve certain aspects of visual-spatial cognitive performance…” since the improvements were task-specific and not uniformly observed across all measures.
-
Another example is the phrase “subjects encountered stressful events” (p. 5, line 143), which is ambiguous. Clarify whether this was self-reported stress and over what timeframe.
-
-
Interpretation of Results:
-
While the paper notes significant improvements in the PFC and MRT post-tests (p. 8, lines 278–280), it would be helpful to discuss why no significant improvements were observed in the MS test. Does this suggest test-specific responsiveness or limitations in the training's scope?
-
This conclusion would benefit from a more balanced discussion of the findings. While positive effects are highlighted, the lack of between-group differences (p. 7, line 276) and the fact that improvements were not exclusive to the HIIT or LVIT conditions should be more explicitly addressed.
-
-
Overstatement and Generalizations
-
Phrases such as “this highlights the potential of integrating structured exercise with cognitive challenges as a promising intervention strategy” (Abstract, p. 1) may overstate the findings given the preliminary nature of the data and small sample size (N=72).
-
Additionally, the conclusion (p. 10) suggests the intervention “may support enhanced performance in tasks requiring rapid decision-making and spatial judgment.” This claim may not be fully supported because such real-life tasks were not directly assessed.
-
-
Presentation of Figures and Tables
-
Ensure that all figures and tables are labeled with complete titles, legends, and appropriate statistical indicators (e.g., p-values, effect sizes, and confidence intervals where relevant). For example, Figures 1–3 lack details on specific significance testing between groups and time points.
-
-
Supplementary Materials:
-
The supplementary file is quite sparse and does not provide sufficient detail to replicate cognitive training. Consider examples or screenshots of actual tasks, detailed timings, or software used.
-
Minor Points
-
There is an inconsistent use of terms such as “HIIT group” and “HITT group” (e.g., Figures 1–3). Ensure consistency across all figures and the text.
-
Abbreviations such as “W1–W4” should be explained clearly at first use.
-
The ethical approval date is 2013 (p. 5, line 144), which may raise questions about the timing of data collection relative to publication. Please clarify whether this study was conducted recently or based on earlier approved protocols.
This was a promising and methodologically well-constructed study. Improvements in language, clarification of findings, and moderation in interpreting results have strong potential for publication and significant relevance to the fields of cognitive health and exercise science.
Comments on the Quality of English LanguageWhile the manuscript generally communicates the research aims and findings effectively, the quality of English requires improvement to ensure clarity, precision, and academic tone throughout. Several issues related to grammar, phrasing, and word choice are present, which could impact the reader's understanding. Below are specific observations:
-
Example: “In the last years physical exercise has been widely associated with cognitive performance…” (p. 3, line 44)
-
Issue: The phrase "in the last years" is awkward.
-
Suggested Correction: “In recent years, physical exercise has been widely associated with cognitive performance…”
-
-
Example: “The inhibitory effects of CT on HIIT or LVIT allowed a significant decrease of the time…” (p. 9, line 339)
-
Issue: The term “inhibitory effects” is confusing in this context and may not reflect the intended meaning.
-
Suggested Correction: Clarify whether the authors mean “interactive effects,” “complementary effects,” or something else.
-
-
Example: “Subjects encountered stressful events before each training session.” (p. 5, line 143)
-
Issue: “Encountered” is vague. It is unclear whether these were reported, measured, or assumed.
-
Suggested Correction: “Participants self-reported experiencing stressful events before the training sessions.”
-
-
Example: “This test was performed on the computer without time limits.” (p. 5, line 191)
-
Issue: The phrasing “on the computer without time limits” is awkward.
-
Suggested Correction: “This computerized test was administered without a time limit.”
-
-
Example: “...the participant was encouraged to pedal as fast as possible against a fixed resistance, set at 75% of the body mass (in Nm).” (p. 6, lines 217–218)
-
Issue: The phrase “fixed resistance, set at 75% of the body mass (in Nm)” is overly technical without sufficient context for a general audience.
-
Suggested Correction: Consider simplifying or explaining what “in Nm” (Newton-meters) signifies in relation to body mass.
-
-
Example: The abbreviation “HITT” is used instead of “HIIT” in several figure captions (e.g., p. 8, lines 282, 286).
-
Issue: Inconsistent use of acronyms can confuse readers.
-
Suggested Correction: Ensure all acronyms are consistent and clearly defined at first use.
-
-
Example: “This highlights the potential of integrating structured exercise with cognitive challenges as a promising intervention strategy for promoting women’s cognitive health…” (Abstract, line 37)
-
Issue: The sentence is long and contains multiple concepts.
-
Suggested Correction: Split into two sentences for clarity: “These findings highlight the potential of combining structured exercise with cognitive challenges. Such interventions may promote cognitive health in women…”
-
Such revisions will significantly enhance readability and ensure that the scientific content is communicated with the clarity and professionalism it deserves.
Author Response
We sincerely thank the reviewers for their time and thoughtful evaluation of our manuscript. We truly appreciate their constructive suggestions, which have helped us improve the quality and clarity of the paper. We are also grateful for their positive comments and encouraging feedback, which we found both motivating and reassuring.

Reviewer 2 Report
Comments and Suggestions for Authors
Major Concerns
- Clarity of the Primary Finding and Hypothesis Support: The central finding is that all groups (including the COG and CTRL groups) improved from pre- to post-test on the PFC and MRT tasks. The manuscript states there were "no significant between-group differences," meaning the improvement was similar regardless of the intervention. This crucial point is somewhat buried and needs to be the focal point of the results and discussion. The hypothesis was that HIIT/LVIT+CT would be "particularly associated with enhanced... effects," which the data do not strongly support. The discussion should be reframed to highlight that the cognitive training itself (or perhaps a practice effect) was the primary driver of improvement, and that the interval training did not provide an additional acute boost beyond that.
- Statistical Analysis Plan: The Methods state a MANOVA was used to analyze the influence of the acute bout on visual-spatial abilities. However, the key outcome is change in performance (Post - Pre). It is unclear if the MANOVA was run on the post-test scores while controlling for pre-test scores (ANCOVA approach) or on the change scores themselves. This must be explicitly detailed. Furthermore, reporting the statistical power only for the significant within-group pre-post comparisons is insufficient. The power for the non-significant between-group comparisons (the test of the main hypothesis) is critical and should be reported or discussed, as a lack of power could explain the null finding.
- Control for Practice Effects: The most plausible explanation for the universal improvement is a practice effect, as the same or very similar tests were administered twice within 48 hours. The cognitive training session itself also consisted of exercises "similar to the twisting and bending exercises used to assess subjects," which would directly prime participants for the post-test. The authors must acknowledge this limitation directly and discuss its implications for interpreting the results. The lack of a pure passive control (e.g., a group that did nothing between pre- and post-test) makes it difficult to isolate the effect of the interventions from simple practice.
Specific Comments/Questions/Suggestions
Title and Abstract:
Title: The title ("Enhancing... Through Acute Interval and Cognitive Training") is slightly overstated as the interval training did not show a unique effect over cognitive training alone. Consider moderating it, e.g., "The Combined Effect of Acute Interval and Cognitive Training on Visual-Spatial Abilities in Women: Preliminary Insights."
Abstract, Lines 18-19: "No significant between-group differences were observed." This is a critical result and should be stated more prominently before detailing the within-group improvements.
Abstract, Line 20: "The time to complete the PFC and MRT tests was significantly lower..." It must be clarified that this was a within-group (pre vs. post) comparison, not a between-group difference.
Introduction:
Lines 60-62: The transition to the study's purpose is excellent, clearly identifying the gap in literature (women, acute IT+CT).
Line 67: The hypothesis could be slightly refined based on the results. It predicted HIIT/LVIT+CT would be "particularly associated" with enhancement. The results suggest it is associated with enhancement, but no more particularly than CT alone.
Methods:
Subjects (Table 1):
The height SD (0.5m, 0.6m) is implausibly large (e.g., 1.7 ± 0.6m implies some women were 2.3m tall). This is likely a typo (probably 0.05m, 0.06m). This must be corrected.
Please define what the "Score PAR" represents (e.g., 0-7 scale, what do the numbers mean?).
Experimental Design:
Line 108: "not to participate in any physical exercise in the weeks in which they were occupied in the experimental protocol." This is a very long period of detraining for "active" women. Please clarify the duration ("the week of the study"?) and justify this requirement.
The design of the COG and CTRL groups is good. However, it is important to note that the COG group received CT and listened to music, while the experimental groups received CT and did physical exercise. The music listening is a constant, well-controlled factor.
Cognitive Performance:
MRT Test (Lines 149-155): The decision to split the test is sound. However, please provide a citation or rationale for the re-proportioning of the time limit from 8 to 4 minutes per block. Was this validated? Could this increased time pressure have influenced results?
Experimental Treatment:
Cognitive Training (Line 175): Stating that "no strategies were suggested" is a strength. Please consider adding a sentence confirming that the experimenter did not provide feedback on performance during the CT session, to rule out instructional confounds.
Statistics:
The statistical plan needs more detail. Specifically:
What were the dependent variables in the MANOVA? (The scores from all three tests? Change scores?).
What were the factors? (Group was a factor, what else?).
Please confirm that menstrual phase, OC use, and stress were included as covariates in the models as stated.
For the significant pre-post comparisons, were paired t-tests used? This should be specified.
Results:
Physiological Data (Table 3):
It is unsurprising that HR was higher in HIIT/LVIT groups. This table could be moved to the supplementary materials to streamline the main results.
Cognitive Performance Data:
Figures 1, 2, 3: The figures are clear. However, the captions must explicitly state that the error bars represent Standard Deviation (SD). The y-axis labels for "No. of responses" in Figures 1 and 2 are confusing – does this mean "Number of Correct Responses"?
The narrative should lead with the primary result: "No significant between-group differences were observed..." before detailing the within-group changes.
For the MRT, you report time decreased significantly (p<0.05) but the number of correct responses (N MRT) does not appear to have changed significantly in any group. This dissociation (faster but not more accurate) should be noted and briefly discussed. Were accuracy and speed analyzed separately?
Discussion:
The discussion is well-written but should be restructured to directly address the main result: comparable improvement across all groups.
Line 236-240: The argument that the improvement "could be due to the development of a new resolution strategy of the cognitive training" is the most likely explanation (i.e., a practice effect). This should be the central point of the discussion, not a secondary point.
Line 240-241: "The inhibitory effects of CT on HIIT or LVIT..." This phrasing is confusing. Do you mean the interactive effects? Or that CT facilitated performance after HIIT/LVIT? Please rephrase for clarity.
Line 241-242: The decrease in time and errors was seen in all groups that received CT (HIIT, LVIT, COG), and to a similar extent. The control group (CTRL) also improved on the PFC test. This strongly points to a practice effect from the cognitive training (and test repetition) as the primary mechanism.
The discussion challenging the inverted-U theory (Lines 225-235) is interesting but should be tempered by the fact that the HIIT group did not perform better than the COG group.
The practical applications (Lines 278-285) are valuable but should be framed as "combined physical and cognitive training" being beneficial, without overstating the unique role of the high-intensity interval component.
Limitations:
The limitations section (Lines 266-275) is good. The following should be added:
Practice Effects: The potential for significant practice effects due to the short interval between pre- and post-testing and the similarity between the CT and assessment tasks.
Lack of Pure Control: The absence of a group that did no training between tests to establish a true baseline for practice effects.
Statistical Power: A discussion of whether the study was sufficiently powered to detect between-group differences, given the null result.
Minor Corrections:
Line 50: "injuries" (spelling).
Line 291 (Fig 2 Caption): "HITT" should be "HIIT".
Throughout: Ensure consistent use of abbreviations after they are defined (e.g., sometimes "RPE" is used, other times "ratings of perceived exertion" is written out).
Data Availability Statement:
The manuscript does not contain a "Data Availability Statement." This is now a standard requirement for most journals. The authors should add a statement indicating whether the data are available in a repository and how to access them, or state that the data are not publicly available due to privacy/ethical restrictions.
Author Response

(The authors gave the same response as above.)

Reviewer 3 Report
Comments and Suggestions for Authors
The manuscript study was to examine the effect of types training combined with cognitive training on visual-spatial performance in young adult women. The topic is presented as original and relevant to the field of Sports Science, as it distinguishes from much of the existing literature, which has focused primarily on young adult male soccer players, middle-aged individuals, or older adults. The methodology of the present study considers instruments appropriate. The manuscript's conclusions are duly solid, reflected with scientific evidence and consistent arguments, with objectivity to the research problem presented. As a suggestion for future studies, it would be interesting to evaluate the body composition and food consumption of the research subjects.
Author Response

(The authors gave the same response as above.)
